# A systematic review of exercise testing in patients with intermittent claudication: A focus on test standardisation and reporting quality in randomised controlled trials of exercise interventions

**Stefan T. Birkett[1], Amy E. Harwood[2], Edward Caldow[3], Saïd Ibeggazene[4], Lee Ingle[5], Sean Pymer[6]***

**1** School of Sport and Health Sciences, University of Central Lancashire, Preston, United Kingdom, **2** Centre for Sports, Exercise and Life Sciences, Coventry University, Coventry, United Kingdom, **3** School of Health and Society, University of Salford, Salford, United Kingdom, **4** College of Health, Wellbeing and Life Sciences, Sheffield Hallam University, Sheffield, United Kingdom, **5** Department of Sport, Health and Exercise Science, University of Hull, Hull, United Kingdom, **6** Academic Vascular Surgical Unit, Hull York Medical School, Hull, United Kingdom

* Sean.Pymer@hey.nhs.uk

## Abstract

A systematic review was conducted to identify the range of terminology used in studies to describe maximum walking distance and the exercise testing protocols, and testing modalities used to measure it in patients with intermittent claudication. A secondary aim was to assess the implementation and reporting of the exercise testing protocols. CINAHL, Medline, EMBASE and Cochrane CENTRAL databases were searched. Randomised controlled trials whereby patients with intermittent claudication were randomised to an exercise intervention were included. The terminology used to describe maximal walking distance was recorded, as was the modality and protocol used to measure it. The implementation and reporting quality was also assessed using pre-specified criteria. Sixty-four trials were included in this review. Maximal walking distance was reported using fourteen different terminologies. Twenty-two different treadmill protocols and three different corridor tests were employed to assess maximal walking distance. No single trial satisfied all the implementation and reporting criteria for an exercise testing protocol. Evidence shows that between-study interpretation is difficult given the heterogenous nature of the exercise testing protocols, test endpoints and terminology used to describe maximal walking distance. This is further compounded by poor test reporting and implementation across studies. Comprehensive guidelines need to be provided to enable a standardised approach to exercise testing in patients with intermittent claudication.

**Data Availability Statement:** All relevant data are within the paper and its Supporting Information files.

**Funding:** The authors received no specific funding for this work.

**Competing interests:** No authors have competing interests.

# Introduction

Peripheral artery disease (PAD) is characterised by atherosclerotic lesions of the arteries in the lower limbs, resulting in a reduction of blood flow [1]. Globally, it is estimated that 236 million people are living with PAD [2]. The classical symptom of PAD is intermittent claudication (IC), characterised by ischaemic muscle pain precipitated by exertion and relieved by rest [3]. IC profoundly decreases walking capacity, physical activity levels, functional ability and leads to poorer quality of life [4]. Supervised exercise therapy is recommended as first line treatment for patients with IC [3, 5], and it is effective for ameliorating symptoms by improving walking capacity and quality of life [6].

To assess change/improvement in functional capacity following an exercise intervention or rehabilitation programme, clinicians or practitioners commonly measure maximal walking distance (MWD). Indeed, this is often the primary outcome to assess the efficacy of treatments in randomised controlled trials (RCT's) [7, 8]. The measurement of MWD involves a patient walking for as long as possible until they are limited by their ischaemic leg symptoms [9]. However, there are a number of different terms used to describe MWD and it is measured using a variety of protocols [8, 9], making direct comparison between studies challenging and results less reproducible. In addition, regardless of testing protocol, procedures are often poorly implemented [10] and it is not always clear whether patients have reached MWD or if the test was terminated for other reasons. These issues regarding unstandardised exercise testing and reporting of outcomes have also been highlighted in a recent scientific statement from the American Heart Association [7]. Therefore, the aims of this systematic review were to: assess the terminology used to describe MWD, examine the various testing protocols used in randomized controlled trials including exercise interventions, and assess the implementation and reporting of exercise testing protocols using adapted recommendations and guidelines for patients with IC [9–13]. These assessments were only made for randomised controlled trials that included an exercise intervention to ensure that there was sufficient scope, but also to ensure that the review had focus, rather than including an unnecessarily large number of trials.

# Methods

## Search strategy and inclusion criteria

We included prospective RCTs in patients with IC where MWD was measured via a clearly specified protocol and patients were part of a structured exercise training programme. A structured exercise training programme was defined as one that stated the prescribed frequency, intensity and / or duration. Studies that included patients with critical limb-threatening ischaemia or asymptomatic PAD were excluded. In addition, studies that randomised patients to an exercise or comparator arm following revascularisation were also excluded. Finally, studies that offered all patients the same exercise programme with or without some other intervention (e.g. exercise + drug therapy vs. exercise alone) were excluded. Exclusion of these studies, and non-RCT's, is in line with a previous review on reporting standards and also ensured that the current review and the number of included studies was manageable [14].

The search was conducted using the following databases: CINAHL, Medline, EMBASE and Cochrane CENTRAL. Terms related to IC, peripheral arterial disease, exercise and MWD (S2 Table). Only full-text articles published from 1995 up to June 2020 in the English language were included. We excluded studies that were published prior to 1995 as the majority of exercise programmes after this date were based on a specific meta-analysis [15] felt that the period of 25 years would provide sufficient information to inform our findings, especially given the increase in exercise-based trials in recent years. In addition, five existing systematic reviews

and meta-analyses were used to identify other trials eligible for inclusion [6, 16–19]. Where authors made reference to published protocols these were also consulted.

## Data management, selection and collection process

Search results were imported into Covidence (Covidence systematic review software, Veritas Health Innovation, Melbourne, Australia) for duplicate removal and screening. Titles and abstracts were screened by two independent reviewers (SB and SP) and conflicts resolved via consensus. The full texts of any potentially eligible articles were then independently screened against the inclusion criteria by the same two reviewers.

Data extraction was performed equally by five reviewers (SB, AEH, SI, EC and SP) using a standardised Microsoft Excel database form (Microsoft, 2010, Redmond, WA, USA). Extraction was then cross-checked for accuracy by two reviewers (SB and SP). Data extraction included study characteristics (author, year, country), sample size and description, a description of the exercise intervention and comparator conditions, length of follow-up and methods for assessing MWD, including how walking distance was reported, the modality, protocol and whether it was the primary or secondary outcome.

## Assessment of walking distance description and implementation

For the purpose of scoring protocols on their reporting standards, for treadmill tests, implementation was evaluated via a modified version of the criteria outlined by Hiatt *et al* [10]. The original criteria were applied when directly observing treadmill tests and as such, certain criteria have been omitted or modified as they cannot be clearly confirmed via reported methods only. This modified version included information about the testing equipment and protocol, pre-test instructions, and the steps involved in conducting the test (Table 1). A maximum score of 11 was available.

One element of the original criteria that could have been reported, 'failure to have the patient straddle the moving belt at the start of the treadmill' has not been included. We felt

**Table 1. Treadmill testing criteria.**

| Criteria | Possible Score |
|---|---|
| **Testing equipment and protocol** | |
| • Clearly states equipment was calibrated | 1 |
| • Cites and correctly implements protocol | 1 |
| **Pre-Test** | |
| • Clearly states participants were rested prior to the test | 1 |
| • Clearly states participants were in a fasted state and instructed to avoid cigarette smoking and alcohol before the test | 1 |
| • States if a claudication pain scale was used | 1 |
| • Maximum walking distance was explained as the sole termination criteria (excluding safety criteria) | 1 |
| **Conducting the test** | |
| • Qualification / skill level of the test administrator is documented | 1 |
| • Clearly states that a familiarisation test to the same protocol was used | 1 |
| • Clearly states that the treadmill screen / clock, timer or watch was hidden from the participant | 1 |
| • Clearly states whether handrail support was permitted | 1 |
| • Clearly states whether all (or what % of) participants terminated the test due to maximally tolerated claudication pain | 1 |
| Maximum possible score: 11 | |

that, based on professional experiences and the balance limitations experienced by those with IC [20], this was not a safe practice and as such this criteria was omitted.

For corridor-based tests, namely the incremental shuttle walk test (ISWT), and six-minute walking test (6MWT) we assessed implementation based on the original study by Singh *et al* [11] and the guidelines provided by the American Thoracic Society (ATS) respectively [13]. We also considered the original PAD specific 6MWT study conducted by Montgomery *et al* [12]. For these corridor tests, a pro forma was used that again included information about the testing equipment and protocol, the pre-test instructions, and the steps involved in conducting the test (Table 2). A maximum score of 10 was available for the 6MWT and 13 for the ISWT. For the 6MWT, there is no equipment that requires calibration, whereas for the ISWT, the audio file used for testing should be calibrated. In addition, the 6MWT is not designed to measure claudication limited MWD, whilst the ISWT is.

For both the treadmill and corridor-based tests, reference to previous studies (including the original studies of Hiatt *et al*, Singh *et al* or Montgomery *et al*) was not sufficient to deem that the same specific methodology had been used and all elements had to be explicitly stated to satisfy our criteria. However, for the 6MWT, reference to the ATS guidelines was deemed sufficient to satisfy our criteria, regardless of whether each element was specifically stated. The rationale for this is that the ATS provide specific practice guidelines [12] whilst other documents simply report outcomes from previous studies. If a study outlined a specific element in its methodology which cited the ATS guidelines, we checked for compliance. If compliance was breached the study was penalised.

**Table 2. Corridor walk testing criteria (6MWT and ISWT).**

| Criteria | Possible Score |
|---|---|
| **Testing equipment and protocol** | |
| • Clearly states equipment was calibrated * | 1 |
| • Cites protocol or accepted guidelines | 1 |
| **Pre-Test** | |
| • The corridor length is stated and in line with the cited protocol / guidelines | 1 |
| • Clearly states that participants were instructed not to perform any vigorous exercise 24hrs before the test | 1 |
| • Clearly states participants were rested prior to the test | 1 |
| • Clearly states participants were instructed to avoid cigarette smoking and alcohol before the test | 1 |
| • States if a claudication pain scale was used | 1 |
| • Clearly states that standardised instructions / requirements about the test were given to the participant | 1 |
| • Maximum walking distance was explained as the sole termination criteria (excluding safety criteria) * | 1 |
| **Conducting the test** | |
| • Qualification / skill level of the test administrator is documented | 1 |
| • Clearly states that a familiarisation test to the same protocol was used | 1 |
| • Clearly states standardised verbal phrases were administered during the test | 1 |
| • Clearly states whether all (or what % of) participants terminated the test due to maximally tolerated claudication pain * | 1 |
| Maximum possible score: 10 (6MWT) or 13 (ISWT) | |

*shuttle walk tests only. 6MWT, six-minute walk test; ISWT, incremental shuttle walk test

## Data analysis

Data analysis regarding how walking distance was reported, and the modality and protocol used to measure it, is presented as number and percentage for each combination. In addition, the number and percentage of studies meeting each implementation criteria is presented narratively and graphically.

## Results

Our search yielded 2836 results. Of these, sixty-four trials, including 3881 patients, met the inclusion criteria and were ultimately included in this review (Fig 1) [21–84]. A list of included trials, including a brief description of the intervention(s) and comparator(s), is provided in S1 Table. MWD was stated as the primary outcome in 55% of trials, though it was not specified in 14%.

### Exercise testing protocols and terminology

Of the sixty-four RCT's that assessed MWD, twenty-nine employed a graded treadmill test [21–24, 27–31, 35–39, 42, 46, 48–52, 56, 57, 63, 68, 74, 77, 82, 83], and twelve a constant work rate treadmill test [26, 34, 41, 47, 53, 66, 67, 69–71, 73, 76]. Three trials conducted both a graded and a constant work rate test [45, 58, 59], fourteen conducted a graded test and a 6MWT [25, 40, 43, 55, 60–62, 64, 65, 72, 75, 78–80], and one employed a constant work rate test and a 6MWT [44]. Of the five trials to only employ a corridor test, two conducted a 6MWT [32, 54], two an incremental shuttle walk (ISWT) [81, 84] and one a modified ISWT [33].

Regarding the exercise testing protocols, forty-six studies that conducted a graded test, eleven different protocols were employed. The most widely adopted protocols were the Gardener-Skinner and Hiatt protocols accounting for 65% and 13%, respectively [85, 86]. From the sixteen trials that adopted a constant work rate test, eleven different protocols were employed, commonly using a speed of 3 km/h at a 10% gradient (four trials, 25%).

The MWD was reported using fourteen different terminologies: peak walking time [21, 22, 38, 42, 51, 63–65, 79], maximal/maximum walking distance [23–25, 27, 33, 34, 41, 46, 55, 70, 71, 73–77, 79–81, 85], absolute claudication distance [26, 35, 44, 52, 61, 62, 66, 67, 69, 71], maximal walking time [28, 29, 36, 37, 39, 40, 47–50, 57, 68, 83], total walking distance [30, 31], total walking capacity [56], time to maximal pain [59, 78], time to exhaustion [82], maximum pain distance [27], absolute walking distance [53], exercise duration [43, 58], maximal claudication pain [60], maximal distance [43], and six minute walk distance [32, 54]. Only two trials documented the level of encouragement given to the patient during treadmill testing. One trial stated encouragement was refrained [34], and the other that it was administered [29].

### Implementation and reporting checklist for treadmill testing

Fig 2 panel A shows the percentage of RCT's satisfying each of the treadmill testing implementation and reporting criteria as shown in Table 1. Full analysis and rating of the individual trials is shown in S3 Table. No single trial satisfied all the criteria. The MWD was stated as the sole test termination criteria in forty-four trials (71%). The remaining 29% of trials incorporated additional criteria (Table 3) such as: volitional exhaustion or fatigue, or reaching a certain time, distance, pain scale or heart rate based endpoints [23, 28, 29, 34, 37, 41–44, 46, 68, 72–74]. Only nine trials (15%) reported whether all (or what % of) patients terminated the test due to claudication limited MWD.

Twenty-six trials (42%) cited and correctly implemented the intended protocol. However, nine trials employed the Gardener-Skinner protocol but failed to cite the original publication

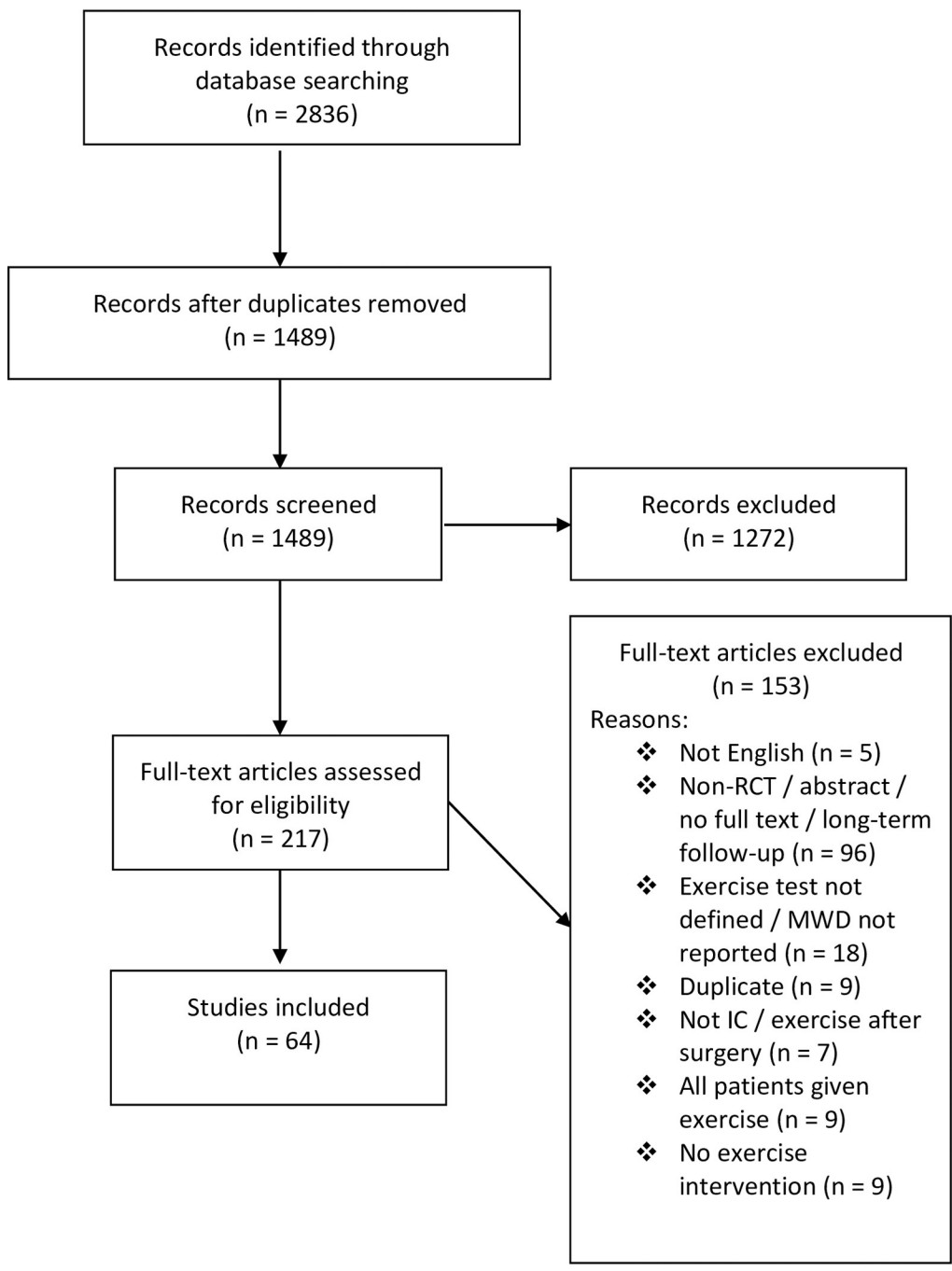

**Fig 1. PRISMA flow chart for study identification.** MWD, maximal walking distance; IC, intermittent claudication.

[21, 43, 55, 59, 63, 68, 72, 78, 79], whilst three trials cited the publication but failed to correctly implement the protocol due to a modification of the speed [23, 40], or incline [80]. As such these variations were counted as three separate graded test protocols. Familiarisation to the treadmill testing protocol was reported in twenty-nine trials (47%). Sixteen trials (26%) reported that a claudication specific pain scale was used during the test, whilst five trials used the CR-10 rating of perceived exertion (Borg) scale to monitor pain [23, 40, 45, 58, 74]. The qualification/skill level of the test administrator was reported in fourteen trials (23%).

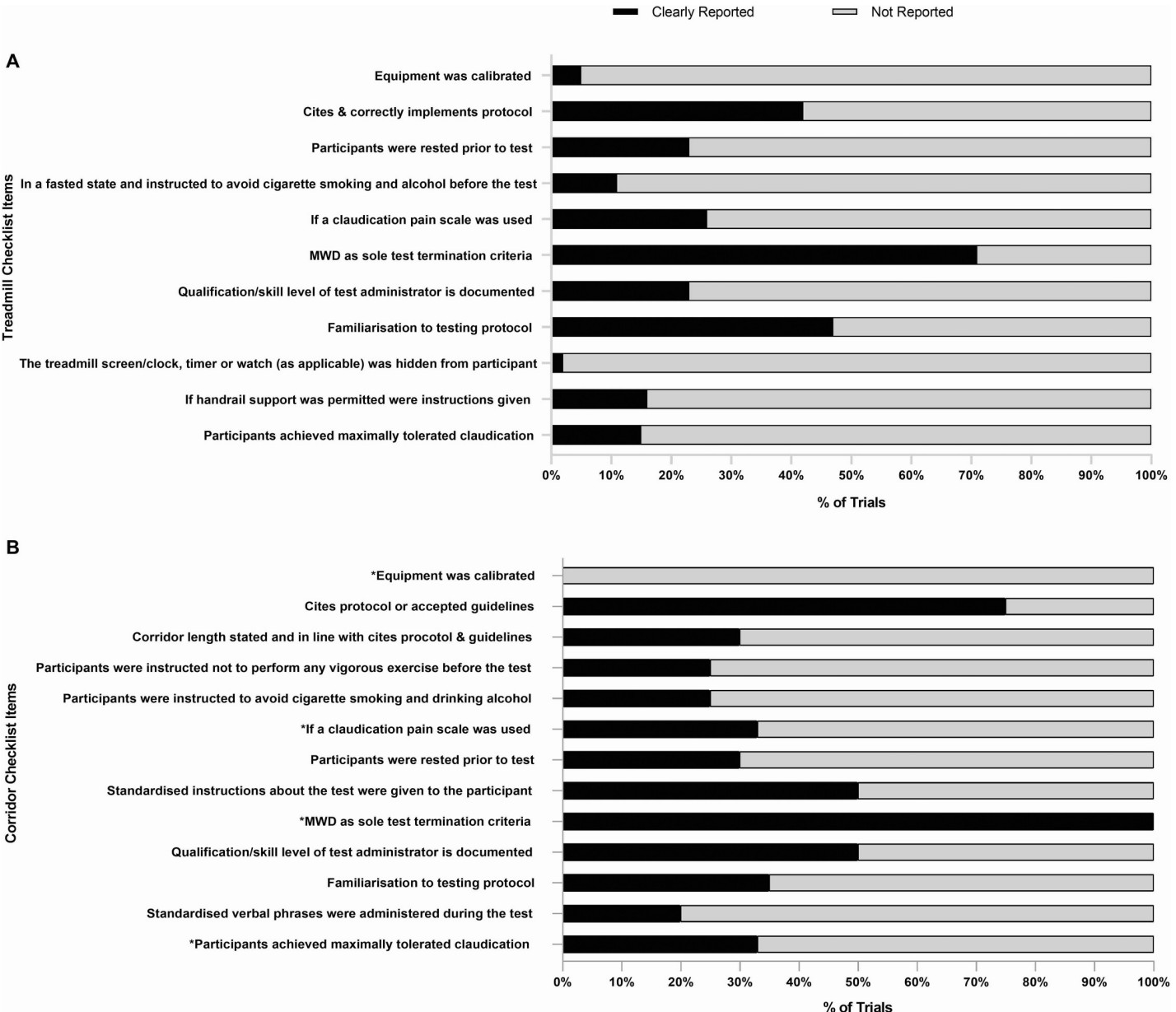

**Fig 2. The percentage of trials that reported the treadmill and corridor testing criteria.** MWD, maximal walking distance.

Seven trials (11%) clearly stated that participants were in a fasted state and instructed to avoid cigarette smoking and drinking alcohol before the test, whilst fourteen (23%) stated that patients were rested prior to the test. A small number of trials reported that the treadmill was calibrated (5%) and that the treadmill screen/clock, timer or watch (as applicable) was hidden from participants during the test (2%). Finally, ten trials (16%) specified whether handrail support was permitted.

## Implementation and reporting checklist for corridor testing

Fig 2 panel B shows the percentage of RCT's satisfying each of the corridor testing implementation and reporting criteria as shown in Table 2. Full analysis and rating of the individual

**Table 3. Additional test endpoints adopted in treadmill tests.**

| Endpoint |
| --- |
| Volitional exhaustion[23, 28, 29, 46, 74] |
| Fatigue and shortness of breath [43] |
| Time limit: 30 min [23, 52], 25 min [29, 46] |
| Distance limit: 207m [41] and 1000m [34] |
| Score of 8 on the CR-10 RPE scale [23] |
| A score of 3 or 4 on the pain scale [68] |
| 80% of heart rate maximum [44] |
| Point of termination [25, 37] |
| Exercise induced factors [42] |
| Strong pain [72] |
| Walking distance in the absence of claudication pain [73] |

trials is shown in S4 Table. No single trial satisfied all the criteria. Fifteen trials (75%) cited a previous protocol or accepted guidelines, whilst five (25%) did not. For the 6MWT, eight trials cited Montgomery *et al* [40, 44, 60–62, 64, 65, 75], four cited the ATS guidelines [25, 43, 79, 80], and one cited Guyatt *et al* [54]. For the ISWT both trials cited Singh *et al* [81, 84]. However only six trials (30%) used the correct corridor length in line with the cited protocol or guidelines [25, 40, 43, 64, 79, 80]. The two trials that employed the ISWT reported that the cones were placed ten meters apart [81, 84] and the trial using a modified ISWT reported a 50-metre figure of eight [33]. Seven trials did not state a corridor length for the 6MWT [32, 55, 61–63, 65, 75], whilst four reported varying lengths of 20 meters [78], 22 meters [44], 33 meters [54] and 50 meters [72].

Five trials (25%) reported that patients had abstained from vigorous activity, cigarette smoking and alcohol prior to the test, whilst six (30%) reported that patients were rested prior to the test. Ten trials (50%) implemented standardised instructions/phrases prior to the test, whilst only four (20%) adopted standardised phrases during it. The qualification/skill level of the test administrator and familiarisation to the testing protocol was reported in ten (50%) and seven trials (35%) respectively.

Implementation criteria specific to the ISWT's showed that no trials reported calibration of the audio recording, one trial (33%) used a claudication pain scale [33] and two trials used the CR-10 rating of perceived exertion (Borg) scale to monitor pain [81, 84]. All trials (100%) reported that claudication limited MWD was the sole test termination criteria, however only one trial reported whether all (or what % of) patients terminated the test due to this [33].

## Discussion

Our review aimed to identify the varied terminology used to describe MWD and examine the different protocols used to measure it, in patients with IC. We also aimed to evaluate the implementation and reporting of these testing protocols. Our findings demonstrated substantial heterogeneity in how MWD is reported and measured across RCT's. Furthermore, the implementation and reporting of the exercise testing protocols was relatively poor.

### Maximum walking capacity and exercise testing protocols

In the current review we found fourteen different terminologies used to describe MWD in just sixty-four studies. This lack of standardised terminology may have a direct impact on patient care as a recent Cochrane review [6], which have informed clinical guidelines, only included

trials that described maximum walking distance or time as an outcome measure. Although other variants of MWD will have been included in the review, it is possible that studies adopting less standard terminology may have been missed during the screening process, especially when the term maximum or maximal is not used in the descriptor (e.g. total walking distance). Consequently, it is possible that such reviews do not encompass the full evidence base. Future trials should adopt recognised terminology and include the term maximal or maximum when describing MWD.

Our findings have also shown that twenty-two different treadmill protocols were across just sixty-four RCT's. The heterogenous nature of treadmill testing is a major concern. Graded and constant work rate tests that differ in speed and gradient result in varying relationships between workload and maximal distance/time [87, 88]. As such the choice of testing protocol will directly influence MWD independent of the exercise intervention. Trials should employ an internationally recognised graded exercise test, given it has the highest reliability across differing severities of IC [85, 87]. However, despite the available evidence, only 72% of trials in the current review employed a graded test. As such, almost 30% adopted a suboptimal testing protocol, meaning that direct between-study comparison may be inappropriate. In addition, pooling studies with different, and less reliable testing protocols may impact upon the effect estimate and should be considered in meta-analyses possibly via sensitivity analysis. Indeed, the implications are that incorrect conclusions may be drawn from the analysis and therefore rule out a potentially viable treatment. Of further concern, the inconsistency of treadmill protocols in clinical trials may translate down to general testing in rehabilitation programmes and impact clinical outcomes for patients in 'usual care'.

A lack of clear guidance for treadmill testing in patients with IC is a likely contributor to the heterogeneity identified. Current guidelines fail to provide adequate details or recommendations for the assessment of MWD [3, 5, 89, 90]. Therefore, as previously recommended [87], guidelines for the assessment of MWD, which advocate for a universal graded test protocol, adopting the reporting criteria outlined in Table 1 are required. This will bring vascular care in line with other clinical specialities such as respiratory medicine, who provide comprehensive testing guidelines [91].

For corridor-based testing the 6MWT was the most commonly employed, usually as a secondary outcome. The 6MWT provides important additional information regarding ambulatory function and it may be a better representation of walking in daily life [8, 92]. However, caution is strongly advised when comparing both the 6MWT and a treadmill test following the same exercise intervention, due to the differing responses [93]. This lack of sensitivity to change may be attributed to the ceiling effect that is evident with the 6MWT [94] and the fact that the test is self-paced, thus reducing standardisation.

## Implementation and reporting checklist for treadmill testing

Our review also demonstrated that implementation of treadmill testing in exercise trials is poorly reported with no single trial satisfying all of our criteria. Although claudication limited MWD was stated as the sole test termination criteria in the majority of trials, almost a third incorporated additional endpoints (e.g. 8 out 10 on the CR-10 scale or 3 or 4 on the pain scale). A major concern is that these endpoints will inherently limit a patient's peak performance and therefore lead to an under or overestimation of their true MWD [9]. The implications of this is that each individual trial could under/overestimate the true effect, which would also be the case for pooled effect estimates. Indeed, in the most recent Cochrane review [6] several studies were included that documented additional test endpoints [22, 28, 46, 74]. Trials should aim to ensure that MWD is the primary test termination point, particularly when the

objective is to measure change in walking capacity. Where patients do not terminate due to maximally tolerated claudication but other endpoints, this should also be reported by authors. However, in the current review 85% of trials did not provide this information.

Furthermore, lack of detail regarding familiarisation testing and handrail support use also limits how the individual study results can be interpreted [85]. This lack of standardisation will may lead to measurement variability with data being less reproducible and less sensitive to the exercise intervention [88].

We also found that less than half of the included trials cited and correctly implemented the intended protocol, which inhibits study replication. Furthermore, three trials cited the Gardener-Skinner graded protocol but deviated from it by manipulating the speed and/or incline. A rationale (for reducing the gradient) was provided by one trial, but it was not appropriate given the reduction for severe IC was in reference to a constant work rate test not a graded [87]. Such deviation means that the reliability of the test is unknown and should not be assumed to be comparable to the original protocol [85], which again impacts upon interpretation. This also highlights a limitation of the peer review process, as important issues such incorrect implementation should have been identified and rectified prior to publication.

## Implementation and reporting checklist for corridor testing

Encouragingly 75% of trials cited a protocol or accepted guidelines for the methodology of their corridor test. For the 6MWT (seventeen trials), 47% cited Montgomery *et al*, 24% the ATS guidelines, and 6% Guyatt *et al*. For the ISWT both trials cited Singh *et al* [81, 84]. Whilst Montgomery *et al* is a PAD specific study it does not offer a standardised approach for conducting the 6MWT. In this study, patients were given verbal encouragement every two minutes, however the content and nature of these phrases is unknown. Conversely, the ATS guidelines provide comprehensive standardised verbal phrases which are delivered more frequently (each minute). A concern is that adopting different protocols between trials may impact upon the results given that encouragement effects performance by over 30 meters [95]. This impact is significant as the estimated minimally clinically important difference (MCID) for the 6MWT is 12–34 meters [96]. Interestingly, despite the ATS guidelines being available since 2002, six trials published after this date still chose to cite Montgomery *et al* [12] or Guyatt *et al* [95].

In addition to different protocols being followed, there were also differences with regards to the corridor lengths used, ranging from 20 to 50 meters, despite guidelines advocating 30 metres [13]. Varying corridor lengths may impact upon the distance walked, as patients have to turn and change direction more or less frequently [97]. One study has demonstrated that patients with chronic lung disease walked nearly 50 meters further when the corridor length was 30 vs. 10 meters [97]. A dearth of evidence exists in other clinical populations, therefore further studies are required to examine the influence of encouragement and corridor length on 6MWT performance in patients with IC.

The incorrect corridor length was also apparent for the ISWT. Both trials that implemented the ISWT cited the original Singh *et al* publication [11], but reported that the cones were placed 10 meters apart, when they should be 9 meters apart, allowing for 0.5 meter turning circle at each end [91]. Although this is only a small difference, it has the potential to be significant given it will underestimate patients walking capacity by 1 meter for every shuttle walked. Another limitation of these trials is that they did not perform a familiarisation test. This is very important given that a learning effect of up to 30 meters is apparent with both tests [91].

This review is not without limitations. Firstly, we did not contact study authors for further details regarding the interventions as this was not feasible given the volume of studies

included. Secondly, we did not hand search the reference lists of included studies though it is unlikely any trials were overlooked given that four databases were used, and five existing systematic reviews and meta-analyses were consulted. Finally, we excluded certain studies, such as those that included exercise performed after revascularisation. However, this ensured that the review was manageable and as 64 trials were still included in spite of this, it is unlikely that including these studies would have altered our findings.

## Conclusion

Evidence shows that between-study interpretation is difficult given the heterogenous nature of the exercise testing protocols, test endpoints and terminology used to describe MWD. We recommend that future trials adopt a standardised approach to exercise testing, implementation and reporting by adopting the minimum reporting criteria outlined in Tables 1 and 2. However, we acknowledge that strict word counts may inhibit this, therefore we recommend two actions; first, for authors to publish a protocol and second, to include this information as S1 to S4 Tables. Importantly, we also strongly recommend that specific guidelines are created to recommend an international, standardised testing procedure to measure MWD using a treadmill in patients with IC, providing step-by-step guidance and standardised reporting terminology.

## Supporting information

**S1 Checklist. PRISMA 2009 checklist.**
(DOC)

**S1 Table. Trial characteristics.**
(DOCX)

**S2 Table. Medline search terms.**
(DOCX)

**S3 Table. Implementation and reporting quality for treadmill testing.**
(DOCX)

**S4 Table. Implementation and reporting quality for corridor testing.**
(DOCX)

**S1 Matrix.**
(XLSX)

## Author Contributions

**Conceptualization:** Stefan T. Birkett, Amy E. Harwood, Edward Caldow, Saïd Ibeggazene, Lee Ingle, Sean Pymer.

**Data curation:** Stefan T. Birkett, Amy E. Harwood, Edward Caldow, Saïd Ibeggazene, Lee Ingle, Sean Pymer.

**Formal analysis:** Stefan T. Birkett, Amy E. Harwood, Edward Caldow, Saïd Ibeggazene, Sean Pymer.

**Methodology:** Stefan T. Birkett, Amy E. Harwood, Edward Caldow, Saïd Ibeggazene.

**Supervision:** Lee Ingle.

**Writing – original draft:** Stefan T. Birkett, Sean Pymer.

**Writing – review & editing:** Stefan T. Birkett, Amy E. Harwood, Edward Caldow, Saïd Ibegga-zene, Lee Ingle, Sean Pymer.

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
