## [Decision Letter · Decision Letter 0]

1 Mar 2021

PONE-D-20-40047

Review of Exercise Testing in Patients with Intermittent Claudication: A Focus on Test Standardisation and Reporting Quality

PLOS ONE

Dear Dr. Pymer,

Thank you for submitting your manuscript to PLOS ONE. After careful consideration, we feel that it has merit but does not fully meet PLOS ONE’s publication criteria as it currently stands. Therefore, we invite you to submit a revised version of the manuscript that addresses the points raised during the review process.

I agree with all reviewer comments and invite authors to respond to each of them.

Authors should further justify why they choose only RCT trials since this is not an interventional review. include other study designs(preferably) or reflect in the title and elsewhere that this is a review of RCTs. Clarify the statement "studies that randomised patients to an exercise or comparator arm following revascularisation without a non-invasive exercise arm, were also excluded" is not quite clear. Provide a better contextualisation of the reason for including only studies published between 1995-2020 given the meta analyses by Gardener and Poehlman, had a different focus from this review. Upgrade the supplementary Table 1 with more specific details focus of the review.

Provide rationale why the review is only focused studies of exercise interventions included in review given that treadmill testing is commonly used in other non-exercise interventions for this population? Fully present the full analysis and rating of the studies included e.g. in table format to enhance transparency and make it easier for the reader to see how each study has been evaluated.

We look forward to receiving your revised manuscript.

Kind regards,

Ukachukwu Okoroafor Abaraogu, BMR PT, MSc, PhD

Academic Editor

PLOS ONE

Journal Requirements:

Reviewers' comments:

Reviewer's Responses to Questions

**Comments to the Author**

1. Is the manuscript technically sound, and do the data support the conclusions?

Reviewer #1: Yes

Reviewer #2: Yes

2. Has the statistical analysis been performed appropriately and rigorously? 

Reviewer #1: Yes

Reviewer #2: Yes

3. Have the authors made all data underlying the findings in their manuscript fully available?

Reviewer #1: Yes

Reviewer #2: Yes

4. Is the manuscript presented in an intelligible fashion and written in standard English?

Reviewer #1: Yes

Reviewer #2: Yes

5. Review Comments to the Author

Reviewer #1: This systematic review aimed to 1) assess the terminology used to describe MWD, 2) examine the various testing protocols used in exercise interventions, and 3) assess the implementation and reporting of exercise testing protocols using adapted recommendations and guidelines for patients with IC. The authors concluded that currently there is significant heterogeneity in terminology, protocols and reporting of exercise testing in trials of people with PAD and IC.

This is the first review of its type and contributes important information that should encourage and enable improvements in the design, implementation and reporting of exercise testing in this population.

The strengths of this manuscript include that it presents a clear introduction and establishment of the rationale/ gap in the literature. Weaknesses of this manuscript include the limited explanation of methodological choices e.g. why only studies of exercise interventions included in review? There is a lack of presentation of full analysis and rating of the studies included e.g. in table format. This would enhance transparency and make it easier for the reader to see how each study has been evaluated.

Major issues-

None.

Minor issues-

Exclusion of studies other than on exercise interventions. Perhaps the authors could review the methods to clearly state the rationale for exclusion of non-exercise intervention RCTs or re-run the search/ include studies of interventions other than exercise. Exercise testing is used in PAD and IC research for multiple interventions and this may allow a more comprehensive summary of the current use of exercise testing in PAD and IC research. If not including/ changing methods then this should be addressed in the limitations section.

Presentation of full data/ analysis of included studies. The authors might consider including a table summarising the findings for all studies. This is currently presented in the text but might be mor informative/ easier for the reader to navigate in table form and collected according to study.

Minors inconsistency issues in use of SWT and ISWT abbreviations on pages 6 and 7.

Repetitions of reference in list – Hiatt et al 2005 is included twice (10 and 20). Perhaps review full list for any other inaccuracies.

Reviewer #2: The manuscript, "Review of Exercise Testing in Patients with Intermittent Claudication: A Focus on Test Standardisation and Reporting Quality" reports an important aspect of PAD assessment. The topic is well justified, well written and interesting.

However, I have a few remarks;

I believe that RCTS of clinical trials or simply trial should also come in the title. And the authors should further justify why they choose only RCT trials since this is not an interventional review. Next, in the inclusion criteria, the statement "studies that randomised patients to an exercise or comparator arm following revascularisation without a non-invasive exercise

arm, were also excluded" is not quite clear. What does the authors mean by non-invasive exercise and of what relevance is it the review?

furthermore, Only full-text articles published from 1995 up to June 2020 were included, and the reason for this was based upon a a meta analyses by Gardener and Poehlman, which has a different focus from this review. You need a better reason.

The supplementary Table 1 is quite scanty and not in line with the focus of the review. Please upgrade this with more specific details.

6. PLOS authors have the option to publish the peer review history of their article (what does this mean?). If published, this will include your full peer review and any attached files.

Reviewer #1: **Yes: **chris seenan

Reviewer #2: **Yes: **Jibril Mohammed

---

## [Author Response · Author response to Decision Letter 0]

5 Mar 2021

Mr Sean Pymer

Academic Vascular Surgical Unit

Hull York Medical School

Hull, UK

Dear Dr. Abaraogu

RE: Manuscript ID: PONE-D-20-40047

We wish to thank the reviewers, Chris Seenan and Jibril Mohammed, for their time and efforts in reviewing our manuscript. You kindly summarised the main points from the reviewers, with no additional comments, so we hope that by responding to the reviewer’s comments, we will also have covered the points you raised.

Reviewer #1:

Comment: Exclusion of studies other than on exercise interventions. Perhaps the authors could review the methods to clearly state the rationale for exclusion of non-exercise intervention RCTs or re-run the search/ include studies of interventions other than exercise. Exercise testing is used in PAD and IC research for multiple interventions, and this may allow a more comprehensive summary of the current use of exercise testing in PAD and IC research. If not including/ changing methods, then this should be addressed in the limitations section.

Response: We absolutely agree with this comment, however, we felt that if we included these studies, the review would have been unmanageable and potentially overwhelming for the reader. We also felt that the addition of these studies would not alter our findings. We have therefore added some additional detail in the methods section “exclusion of these studies, and non-RCT's, is in line with a previous review on reporting standards and also ensured that the current review and the number of included studies was manageable” and the limitations section as suggested, “finally, we excluded certain studies, such as those that included exercise performed after revascularisation. However, this ensured that the review was manageable and as 64 trials were still included in spite of this, it is unlikely that including these studies would have altered our findings”. Many thanks for this suggestion.

Comment: Presentation of full data/ analysis of included studies. The authors might consider including a table summarising the findings for all studies. This is currently presented in the text but might be more informative/ easier for the reader to navigate in table form and collected according to study.

Response: Yes, we agree, and clarity is important. We have added the analysis and scoring of included studies in a table format. However, given this created seven new tables we have added these as supplementary materials (tables 3 and 4). 

Comment: Minor inconsistency issues in use of SWT and ISWT abbreviations on pages 6 and 7.

Response: Many thanks for spotting this mistake, we have amended the manuscript accordingly. 

Comment: Repetitions of reference in list – Hiatt et al 2005 is included twice (10 and 20). Perhaps review full list for any other inaccuracies

Response: This has now been rectified. Thankyou. 

Reviewer #2:

Comment: I believe that RCTS of clinical trials or simply trial should also come in the title. And the authors should further justify why they choose only RCT trials since this is not an interventional review.

Response: Many thanks for this comment. We have now updated the title to “Review of Exercise Testing in Patients with Intermittent Claudication: A Focus on Test Standardisation and Reporting Quality in Randomised Controlled Trials”. We chose to exclude non-RCT’s in line with a previous review of reporting quality published in PLOS One (1). We also felt that this would ensure that the review was manageable and not overwhelming for the reader. Finally, standardised reporting is of the greatest importance in RCT’s as these often inform clinical practice or are used to generate level 1a evidence which does inform clinical practice. We have added the following sentence to the methods section “exclusion of these studies, and non-RCT's, is in line with a previous review on reporting standards and also ensured that the current review and the number of included studies was manageable”.

Comment: Next, in the inclusion criteria, the statement "studies that randomised patients to an exercise or comparator arm following revascularisation without a non-invasive exercise arm, were also excluded" is not quite clear. What does the authors mean by non-invasive exercise and of what relevance is it the review?

Response: We appreciate this comment and agree that this sentence is not as clear as it could be. To clarify, by non-invasive exercise arm, we meant an initial randomisation to exercise therapy or revascularisation, rather than post-revascularisation randomisation (i.e. randomised to revascularisation or exercise, rather than to exercise or control after revascularisation). We have edited this sentence to state “studies that randomised patients to an exercise or comparator arm following revascularisation were also excluded"

Comment: Furthermore, only full-text articles published from 1995 up to June 2020 were included, and the reason for this was based upon a meta analyses by Gardener and Poehlman, which has a different focus from this review. You need a better reason.

Response: We appreciate this comment, but our aim was to ensure that the review was manageable and not overwhelming to the reader. We felt that the 25-year period was sufficient especially given the increase in exercise-based trials in recent years. This is demonstrated in our results whereby we still included 64 studies. We have also added the following to the methods section “we excluded studies that were published prior to 1995 as the majority of exercise programmes after this date were based on a specific meta-analysis (14) and we felt that the period of 25 years would provide sufficient information to inform our findings, especially given the increase in exercise-based trials in recent years”. 

Comment: The supplementary Table 1 is quite scanty and not in line with the focus of the review. Please upgrade this with more specific details.

Response: We have now added specific details regarding the exercise intervention and control (when available). Thankyou

We appreciate the opportunity to revise our manuscript and again wish to thank the reviewers and yourself.

Yours Sincerely, 

Mr Sean Pymer – on behalf of all authors

Reference:

1. Tew GA, Brabyn S, Cook L, Peckham E. The Completeness of Intervention Descriptions in Randomised Trials of Supervised Exercise Training in Peripheral Arterial Disease. PLOS ONE. 2016;11(3):e0150869.

---

## [Editor Report · Decision Letter 1]

16 Mar 2021

A Systematic Review of Exercise Testing in Patients with Intermittent Claudication: A Focus on Test Standardisation and Reporting Quality in Randomised Controlled Trials

PONE-D-20-40047R1

Dear Dr. Pymer,

We’re pleased to inform you that your manuscript has been judged scientifically suitable for publication and will be formally accepted for publication once it meets all outstanding technical requirements.

Kind regards,

Ukachukwu Okoroafor Abaraogu, BMR PT, MSc, PhD

Academic Editor

PLOS ONE

Additional Editor Comments (optional):

Congratulation on the revision. I am happy to 'conditionally' accept the manuscript but will like a further very minor revison before the manuscript goes to the production stage

Therefore, I will like the authors to specify in the title and also give a few sentenses in the introcuction to point to te fact that only exercise intervention were reviewed. I agree that the review results may or may not be different when other interventions and research design are included. It is only cautionary to define and describe review title, aim, methods, and findings within the limits of the review scope and the empirical evidence generated.
---

## [Editor Report · Acceptance letter]

22 Mar 2021

PONE-D-20-40047R1 

A Systematic Review of Exercise Testing in Patients with Intermittent Claudication: A Focus on Test Standardisation and Reporting Quality in Randomised Controlled Trials Including Exercise Interventions 

Dear Dr. Pymer:

I'm pleased to inform you that your manuscript has been deemed suitable for publication in PLOS ONE. Congratulations! Your manuscript is now with our production department. 

Kind regards, 

on behalf of

Dr. Ukachukwu Okoroafor Abaraogu 

Academic Editor

PLOS ONE